# SLANG: Fast Structured Covariance Approximations for Bayesian Deep Learning with Natural Gradient

**Aaron Mishkin**[*]
University of British Columbia
Vancouver, Canada
amishkin@cs.ubc.ca

**Frederik Kunstner**[*]
Ecole Polytechnique Fédérale de Lausanne
Lausanne, Switzerland
frederik.kunstner@epfl.ch

**Didrik Nielsen**
RIKEN Center for AI Project
Tokyo, Japan
didrik.nielsen@riken.jp

**Mark Schmidt**
University of British Columbia
Vancouver, Canada
schmidtm@cs.ubc.ca

**Mohammad Emtiyaz Khan**
RIKEN Center for AI Project
Tokyo, Japan
emtiyaz.khan@riken.jp

## Abstract

Uncertainty estimation in large deep-learning models is a computationally challenging task, where it is difficult to form even a Gaussian approximation to the posterior distribution. In such situations, existing methods usually resort to a diagonal approximation of the covariance matrix despite the fact that these matrices are known to result in poor uncertainty estimates. To address this issue, we propose a new stochastic, low-rank, approximate natural-gradient (SLANG) method for variational inference in large, deep models. Our method estimates a "diagonal plus low-rank" structure based solely on back-propagated gradients of the network log-likelihood. This requires strictly less gradient computations than methods that compute the gradient of the whole variational objective. Empirical evaluations on standard benchmarks confirm that SLANG enables faster and more accurate estimation of uncertainty than mean-field methods, and performs comparably to state-of-the-art methods.

## 1 Introduction

Deep learning has had enormous recent success in fields such as speech recognition and computer vision. In these problems, our goal is to predict well and we are typically less interested in the uncertainty behind the predictions. However, deep learning is now becoming increasingly popular in applications such as robotics and medical diagnostics, where accurate measures of uncertainty are crucial for reliable decisions. For example, uncertainty estimates are important for physicians who use automated diagnosis systems to choose effective and safe treatment options. Lack of such estimates may lead to decisions that have disastrous consequences.

The goal of Bayesian deep learning is to provide uncertainty estimates by integrating over the posterior distribution of the parameters. Unfortunately, the complexity of deep learning models makes

---

[*]Equal contributions. This work was conducted during an internship at the RIKEN Center for AI project.

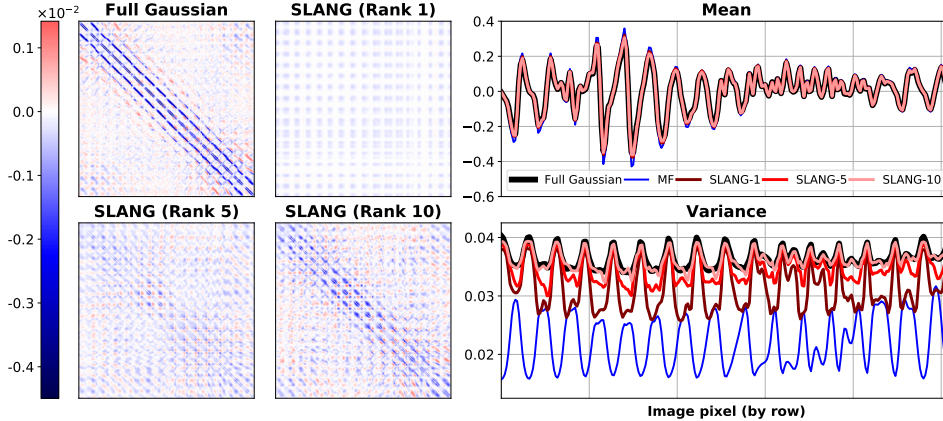

Figure 1: This figure illustrates the advantages of SLANG method over mean-field approaches on the USPS dataset (see Section 4.1 for experimental details). The figure on the left compares our structured covariance approximation with the one obtained by a full Gaussian approximation. For clarity, only off-diagonal entries are shown. We clearly see that our approximation becomes more accurate as the rank is increased. The figures on the right compare the means and variances (the diagonal of the covariance). The means match closely for all methods, but the variance is heavily underestimated by the mean-field method. SLANG's covariance approximations do not suffer form this problem, which is likely due to the off-diagonal structure it learns.

it infeasible to perform the integration exactly. Sampling methods such as stochastic-gradient Markov chain Monte Carlo [9] have been applied to deep models, but they usually converge slowly. They also require a large memory to store the samples and often need large preconditioners to mix well [2, 5, 32]. In contrast, variational inference (VI) methods require much less memory and can scale to large problems by exploiting stochastic gradient methods [7, 12, 28]. However, they often make crude simplifications, like the mean-field approximation, to reduce the memory and computation cost. This can result in poor uncertainty estimates [36]. Fast and accurate estimation of uncertainty for large models remains a challenging problem in Bayesian deep learning.

In this paper, we propose a new variational inference method to estimate Gaussian approximations with a diagonal plus low-rank covariance structure. This gives more accurate and flexible approximations than the mean-field approach. Our method also enables fast estimation by using an approximate natural-gradient algorithm that builds the covariance estimate solely based on the back-propagated gradients of the network log-likelihood. We call our method *stochastic low-rank approximate natural-gradient* (SLANG). SLANG requires strictly less gradient computations than methods that require gradients of the variational objective obtained using the reparameterization trick [24, 26, 35]. Our empirical comparisons demonstrate the improvements obtained over mean-field methods (see Figure 1 for an example) and show that SLANG gives comparable results to the state-of-the-art on standard benchmarks.

The code to reproduce the experimental results in this paper is available at `https://github.com/aaronpmishkin/SLANG`.

## 1.1 Related Work

Gaussian variational distributions with full covariance matrices have been used extensively for shallow models [6, 8, 16, 22, 24, 30, 34, 35]. Several efficient ways of computing the full covariance matrix are discussed by Seeger [31]. Other works have considered various structured covariance approximations, based on the Cholesky decomposition [8, 35], sparse covariance matrices [34] and low-rank plus diagonal structure [6, 26, 30]. Recently, several works [24, 26] have applied stochastic gradient descent on the variational objective to estimate such a structure. These methods often employ an adaptive learning rate method, such as Adam or RMSprop, which increases the memory cost. All of these methods have only been applied to shallow models, and it remains unclear how they will perform (and whether they can be adapted) for deep models. Moreover, a natural-gradient method is preferable to gradient-based methods when optimizing the parameters of a distribution [3, 15, 18].

Our work shows that a natural-gradient method not only has better convergence properties, but also has lower computation and memory cost than gradient-based methods.

For deep models, a variety of methods have been proposed based on mean-field approximations. These methods optimize the variational objective using stochastic-gradient methods and differ from each other in the way they compute those gradients [7, 12, 14, 19, 28, 38]. They all give poor uncertainty estimates in the presence of strong posterior correlations and also shrink variances [36]. SLANG is designed to add extra covariance structure and ensure better performance than mean-field approaches.

A few recent works have explored structured covariance approximations for deep models. In [38], the Kronecker-factored approximate curvature (K-FAC) method is applied to perform approximate natural-gradient VI. Another recent work has applied K-FAC to find a Laplace approximation [29]. However, the Laplace approximation can perform worse than variational inference in many scenarios, e.g., when the posterior distribution is not symmetric [25]. Other types of approximation methods include Bayesian dropout [10] and methods that use matrix-variate Gaussians [21, 33]. All of these approaches make structural assumptions that are different from our low-rank plus diagonal structure. However, similarly to our work, they provide new ways to improve the speed and accuracy of uncertainty estimation in deep learning.

## 2  Gaussian Approximation with Natural-Gradient Variational Inference

Our goal is to estimate the uncertainty in deep models using Bayesian inference. Given $N$ data examples $\mathcal{D} = \{\mathcal{D}_i\}_{i=1}^N$, a Bayesian version of a deep model can be specified by using a likelihood $p(\mathcal{D}_i|\boldsymbol{\theta})$ parametrized by a deep network with parameters $\boldsymbol{\theta} \in \mathbb{R}^D$ and a prior distribution $p(\boldsymbol{\theta})$. For simplicity, we assume that the prior is a Gaussian distribution, such as an isotropic Gaussian $p(\boldsymbol{\theta}) \sim \mathcal{N}(0, (1/\lambda)\mathbf{I})$ with the scalar precision parameter $\lambda > 0$. However, the methods presented in this paper can easily be modified to handle many other types of prior distributions. Given such a model, Bayesian approaches compute an estimate of uncertainty by using the posterior distribution: $p(\boldsymbol{\theta}|\mathcal{D}) = p(\mathcal{D}|\boldsymbol{\theta})p(\boldsymbol{\theta})/p(\mathcal{D})$. This requires computation of the *marginal likelihood* $p(\mathcal{D}) = \int p(\mathcal{D}|\boldsymbol{\theta})p(\boldsymbol{\theta})d\boldsymbol{\theta}$, which is a high-dimensional integral and difficult to compute.

Variational inference (VI) simplifies the problem by approximating $p(\boldsymbol{\theta}|\mathcal{D})$ with a distribution $q(\boldsymbol{\theta})$. In this paper, our focus is on obtaining approximations that have a Gaussian form, i.e., $q(\boldsymbol{\theta}) = \mathcal{N}(\boldsymbol{\theta}|\boldsymbol{\mu}, \boldsymbol{\Sigma})$ with mean $\boldsymbol{\mu}$ and covariance $\boldsymbol{\Sigma}$. The parameters $\boldsymbol{\mu}$ and $\boldsymbol{\Sigma}$ are referred to as the *variational parameters* and can be obtained by maximizing a lower bound on $p(\mathcal{D})$ called the evidence lower bound (ELBO),

$$\text{ELBO:} \quad \mathcal{L}(\boldsymbol{\mu}, \boldsymbol{\Sigma}) := \mathbb{E}_q\left[\log p(\mathcal{D}|\boldsymbol{\theta})\right] - \mathbb{D}_{KL}[q(\boldsymbol{\theta}) \,\|\, p(\boldsymbol{\theta})]. \quad (1)$$

where $\mathbb{D}_{KL}[\cdot]$ denotes the Kullback-Leibler divergence.

A straightforward and popular approach to optimize $\mathcal{L}$ is to use stochastic gradient methods [24, 26, 28, 35]. However, natural-gradients are preferable when optimizing the parameters of a distribution [3, 15, 18]. This is because natural-gradient methods perform optimization on the Riemannian manifold of the parameters, which can lead to a faster convergence when compared to gradient-based methods. Typically, natural-gradient methods are difficult to implement, but many easy-to-implement updates have been derived in recent works [15, 17, 19, 38]. We build upon the approximate natural-gradient method proposed in [19] and modify it to estimate structured covariance-approximations.

Specifically, we extend the Variational Online Gauss-Newton (VOGN) method [19]. This method uses the following update for $\boldsymbol{\mu}$ and $\boldsymbol{\Sigma}$ (a derivation is in Appendix A),

$$\boldsymbol{\mu}_{t+1} = \boldsymbol{\mu}_t - \alpha_t \boldsymbol{\Sigma}_{t+1}\left[\hat{\mathbf{g}}(\boldsymbol{\theta}_t) + \lambda\boldsymbol{\mu}_t\right], \quad \boldsymbol{\Sigma}_{t+1}^{-1} = (1 - \beta_t)\boldsymbol{\Sigma}_t^{-1} + \beta_t\left[\hat{\mathbf{G}}(\boldsymbol{\theta}_t) + \lambda\mathbf{I}\right], \quad (2)$$

$$\text{with } \hat{\mathbf{g}}(\boldsymbol{\theta}_t) := -\frac{N}{M}\sum_{i \in \mathcal{M}} \mathbf{g}_i(\boldsymbol{\theta}_t), \text{ and } \hat{\mathbf{G}}(\boldsymbol{\theta}_t) := -\frac{N}{M}\sum_{i \in \mathcal{M}} \mathbf{g}_i(\boldsymbol{\theta}_t)\mathbf{g}_i(\boldsymbol{\theta}_t)^\top,$$

where $t$ is the iteration number, $\alpha_t, \beta_t > 0$ are learning rates, $\boldsymbol{\theta}_t \sim \mathcal{N}(\boldsymbol{\theta}|\boldsymbol{\mu}_t, \boldsymbol{\Sigma}_t)$, $\mathbf{g}_i(\boldsymbol{\theta}_t) := \nabla_\theta \log p(\mathcal{D}_i|\boldsymbol{\theta}_t)$ is the back-propagated gradient obtained on the $i$'th data example, $\hat{\mathbf{G}}(\boldsymbol{\theta}_t)$ is an Empirical Fisher (EF) matrix, and $\mathcal{M}$ is a minibatch of $M$ data examples. This update is an approximate natural-gradient update obtained by using the EF matrix as an approximation of the

Hessian [23] in a method called Variational Online Newton (VON) [19]. This is explained in more detail in Appendix A. As discussed in [19], the VOGN method is an approximate Natural-gradient update which may not have the same properties as the exact natural-gradient update. However, an advantage of the update (2) is that it only requires back-propagated gradients, which is a desirable feature when working with deep models.

The update (2) is computationally infeasible for large deep models because it requires the storage and inversion of the $D \times D$ covariance matrix. Storage takes $O(D^2)$ memory space and inversion requires $O(D^3)$ computations, which makes the update very costly to perform for large models. We cannot form $\mathbf{\Sigma}$ or invert it when $D$ is in millions. Mean-field approximations avoid this issue by restricting $\mathbf{\Sigma}$ to be a diagonal matrix, but they often give poor Gaussian approximations. Our idea is to estimate a low-rank plus diagonal approximation of $\mathbf{\Sigma}$ that reduces the computational cost while preserving some off-diagonal covariance structure. In the next section, we propose modifications to the update (2) to obtain a method whose time and space complexity are both linear in $D$.

## 3   Stochastic, Low-rank, Approximate Natural-Gradient (SLANG) Method

Our goal is to modify the update (2) to obtain a method whose time and space complexity is linear in $D$. We propose to approximate the inverse of $\mathbf{\Sigma}_t$ by a "low-rank plus diagonal" matrix:

$$\mathbf{\Sigma}_t^{-1} \approx \hat{\mathbf{\Sigma}}_t^{-1} := \mathbf{U}_t\mathbf{U}_t^\top + \mathbf{D}_t, \tag{3}$$

where $\mathbf{U}_t$ is a $D \times L$ matrix with $L \ll D$ and $\mathbf{D}_t$ is a $D \times D$ diagonal matrix. The cost of storing and inverting this matrix is linear in $D$ and reasonable when $L$ is small. We now derive an update for $\mathbf{U}_t$ and $\mathbf{D}_t$ such that the resulting $\hat{\mathbf{\Sigma}}_{t+1}^{-1}$ closely approximates the update shown in (2). We start by writing an approximation to the update of $\mathbf{\Sigma}_{t+1}^{-1}$ where we replace covariance matrices by their structured approximations:

$$\hat{\mathbf{\Sigma}}_{t+1}^{-1} := \mathbf{U}_{t+1}\mathbf{U}_{t+1}^\top + \mathbf{D}_{t+1} \approx (1 - \beta_t)\hat{\mathbf{\Sigma}}_t^{-1} + \beta_t \left[ \hat{\mathbf{G}}(\boldsymbol{\theta}_t) + \lambda\mathbf{I} \right] \tag{4}$$

This update cannot be performed exactly without potentially increasing the rank of the low-rank component $\mathbf{U}_{t+1}$, since the structured components on the right hand side are of rank at most $L + M$, where $M$ is the size of the minibatch. This is shown in (5) below where we have rearranged the left hand side of (4) as the sum of a structured component and a diagonal component. To obtain a rank $L$ approximation to the left hand side of (5), we propose to approximate the structured component by an eigenvalue decomposition as shown in (6) below,

$$(1 - \beta_t)\hat{\mathbf{\Sigma}}_t^{-1} + \beta_t \left[ \hat{\mathbf{G}}(\boldsymbol{\theta}_t) + \lambda\mathbf{I} \right] = \underbrace{(1 - \beta_t)\mathbf{U}_t\mathbf{U}_t^\top + \beta_t\hat{\mathbf{G}}(\boldsymbol{\theta}_t)}_{\text{Rank at most } L + M} + \underbrace{(1 - \beta_t)\mathbf{D}_t + \beta_t\lambda\mathbf{I}}_{\text{Diagonal component}}, \tag{5}$$

$$\approx \underbrace{\mathbf{Q}_{1:L}\mathbf{\Lambda}_{1:L}\mathbf{Q}_{1:L}^\top}_{\text{Rank } L \text{ approximation}} + \underbrace{(1 - \beta_t)\mathbf{D}_t + \beta_t\lambda\mathbf{I}}_{\text{Diagonal component}}, \tag{6}$$

where $\mathbf{Q}_{1:L}$ is a $D \times L$ matrix containing the first $L$ leading eigenvectors of $(1 - \beta_t)\mathbf{U}_t\mathbf{U}_t^\top + \beta_t\hat{\mathbf{G}}(\boldsymbol{\theta}_t)$ and $\mathbf{\Lambda}_{1:L}$ is an $L \times L$ diagonal matrix containing the corresponding eigenvalues. Figure 2 visualizes the update from (5) to (6).

The low-rank component $\mathbf{U}_{t+1}$ can now be updated to mirror the low-rank component of (6),

$$\mathbf{U}_{t+1} = \mathbf{Q}_{1:L}\mathbf{\Lambda}_{1:L}^{1/2},, \tag{7}$$

and the diagonal $\mathbf{D}_{t+1}$ can be updated to match the diagonal of the left and right sides of (4), i.e.,

$$\text{diag}\left[ \mathbf{U}_{t+1}\mathbf{U}_{t+1}^\top + \mathbf{D}_{t+1} \right] = \text{diag}\left[ (1 - \beta_t)\mathbf{U}_t\mathbf{U}_t^\top + \beta_t\hat{\mathbf{G}}(\boldsymbol{\theta}_t) + (1 - \beta_t)\mathbf{D}_t + \beta_t\lambda\mathbf{I} \right], \tag{8}$$

This gives us the following update for $\mathbf{D}_{t+1}$ using a *diagonal correction* $\Delta_t$,

$$\mathbf{D}_{t+1} = (1 - \beta)\mathbf{D}_t + \beta_t\lambda\mathbf{I} + \Delta_t, \tag{9}$$

$$\Delta_t = \text{diag}\left[ (1 - \beta)\mathbf{U}_t\mathbf{U}_t^\top + \beta_t\hat{\mathbf{G}}(\boldsymbol{\theta}_t) - \mathbf{U}_{t+1}\mathbf{U}_{t+1}^\top \right]. \tag{10}$$

This step is cheap since computing the diagonal of the EF matrix is linear in $D$.

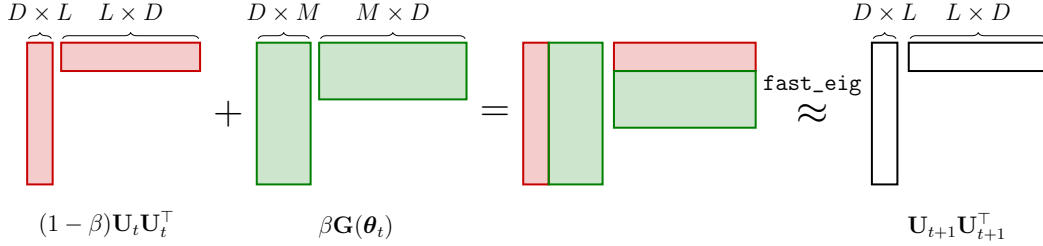

$$\underbrace{D \times L}_{} \ \underbrace{L \times D}_{} \qquad \underbrace{D \times M}_{} \ \underbrace{M \times D}_{} \qquad\qquad\qquad \underbrace{D \times L}_{} \ \underbrace{L \times D}_{}$$

$$(1-\beta)\mathbf{U}_t\mathbf{U}_t^\top \qquad\qquad \beta\mathbf{G}(\boldsymbol{\theta}_t) \qquad\qquad\qquad\qquad \mathbf{U}_{t+1}\mathbf{U}_{t+1}^\top$$

Figure 2: This figure illustrates Equations (6) and (7) which are used to derive SLANG.

The new covariance approximation can now be used to update $\boldsymbol{\mu}_{t+1}$ according to (2) as shown below:

$$\text{SLANG:} \quad \boldsymbol{\mu}_{t+1} = \boldsymbol{\mu}_t - \alpha_t \left[ \mathbf{U}_{t+1}\mathbf{U}_{t+1}^\top + \mathbf{D}_{t+1} \right]^{-1} \left[ \hat{\mathbf{g}}(\boldsymbol{\theta}_t) + \lambda\boldsymbol{\mu}_t \right], \tag{11}$$

The above update uses a stochastic, low-rank covariance estimate to approximate natural-gradient updates, which is why we use the name SLANG.

When $L = D$, $\mathbf{U}_{t+1}\mathbf{U}_{t+1}^\top$ is full rank and SLANG is identical to the approximate natural-gradient update (2). When $L < D$, SLANG produces matrices $\hat{\boldsymbol{\Sigma}}_t^{-1}$ with diagonals matching (2) at every iteration. The diagonal correction ensures that no diagonal information is lost during the low-rank approximation of the covariance. A formal statement and proof is given in Appendix D.

We also tried an alternative method where $\mathbf{U}_{t+1}$ is learned using an exponential moving-average of the eigendecompositions of $\hat{\mathbf{G}}(\boldsymbol{\theta})$. This previous iteration of SLANG is discussed in Appendix B, where we show that it gives worse results than the SLANG update.

Next, we give implementation details of SLANG.

## 3.1 Details of the SLANG Implementation

The pseudo-code for SLANG is shown in Algorithm 1 in Figure 3.

At every iteration, we generate a sample $\boldsymbol{\theta}_t \sim \mathcal{N}(\boldsymbol{\theta}|\boldsymbol{\mu}_t, \mathbf{U}_t\mathbf{U}_t^\top + \mathbf{D}_t)$. This is implemented in line 4 of Algorithm 1 using the subroutine `fast_sample`. Pseudocode for this subroutine is given in Algorithm 3. This function uses the Woodbury identity and to compute the square-root matrix $\mathbf{A}_t = \left( \mathbf{U}_t\mathbf{U}_t^\top + \mathbf{D}_t \right)^{-1/2}$ [4]. The sample is then computed as $\boldsymbol{\theta}_t = \boldsymbol{\mu}_t + \mathbf{A}_t\boldsymbol{\epsilon}$, where $\boldsymbol{\epsilon} \sim \mathcal{N}(\mathbf{0}, \mathbf{I})$. The function `fast_sample` requires computations in $O(DL^2 + DLS)$ to generate $S$ samples, which is linear in $D$. More details are given in Appendix C.4.

Given a sample, we need to compute and store all the individual stochastic gradients $\mathbf{g}_i(\boldsymbol{\theta}_t)$ for all examples $i$ in a minibatch $\mathcal{M}$. The standard back-propagation implementation does not allow this. We instead use a version of the backpropagation algorithm outlined in a note by Goodfellow [11], which enables efficient computation of the gradients $\hat{\mathbf{g}}_i(\boldsymbol{\theta}_t)$. This is shown in line 6 of Algorithm 1, where a subroutine `backprop_goodfellow` is used (see details of this subroutine in Appendix C.1).

In line 7, we compute the eigenvalue decomposition of $(1 - \beta_t)\mathbf{U}_t\mathbf{U}_t + \beta_t\hat{\mathbf{G}}(\boldsymbol{\theta}_t)$ by using the `fast_eig` subroutine. The subroutine `fast_eig` implements a randomized eigenvalue decomposition method discussed in [13]. It computes the top-$L$ eigendecomposition of a low-rank matrix in $O(DLMS + DL^2)$. More details on the subroutine are given in Appendix C.2. The matrix $\mathbf{U}_{t+1}$ and $\mathbf{D}_{t+1}$ are updated using the eigenvalue decomposition in lines 8, 9 and 10.

In lines 11 and 12, we compute the update vector $[\mathbf{U}_{t+1}\mathbf{U}_{t+1}^\top + \mathbf{D}_{t+1}]^{-1}[\hat{\mathbf{g}}(\boldsymbol{\theta}_t) + \lambda\boldsymbol{\mu}_t]$, which requires solving a linear system. We use the subroutine `fast_inverse` shown in Algorithm 2. This subroutine uses the Woodbury identity to efficiently compute the inverse with a cost $O(DL^2)$. More details are given in Appendix C.3. Finally, in line 13, we update $\boldsymbol{\mu}_{t+1}$.

The overall computational complexity of SLANG is $O(DL^2 + DLMS)$ and its memory cost is $O(DL + DMS)$. Both are linear in $D$ and $M$. The cost is quadratic in $L$, but since $L \ll D$ (e.g., 5

| **Algorithm 1** SLANG | **Algorithm 2** fast_inverse$(\mathbf{g}, \mathbf{U}, \mathbf{d})$ |
|---|---|
| **Require:** Data $\mathcal{D}$, hyperparameters $M, L, \lambda, \alpha, \beta$ | 1: $\mathbf{A} \leftarrow (\mathbf{I}_L + \mathbf{U}^\top \mathbf{d}^{-1} \mathbf{U})^{-1}$ |
| 1: Initialize $\boldsymbol{\mu}, \mathbf{U}, \mathbf{d}$ | 2: $\mathbf{y} \leftarrow \mathbf{d}^{-1}\mathbf{g} - \mathbf{d}^{-1}\mathbf{U}\mathbf{A}\mathbf{U}^\top \mathbf{d}^{-1}\mathbf{g}$ |
| 2: $\delta \leftarrow (1 - \beta)$ | 3: **return y** |
| 3: **while** not converged **do** | |
| 4: $\quad \boldsymbol{\theta} \leftarrow$ fast_sample$(\boldsymbol{\mu}, \mathbf{U}, \mathbf{d})$ | |
| 5: $\quad \mathcal{M} \leftarrow$ sample a minibatch | |
| 6: $\quad [\mathbf{g}_1, .., \mathbf{g}_M] \leftarrow$ backprop_goodfellow$(\mathcal{D}_{\mathcal{M}}, \boldsymbol{\theta})$ | **Algorithm 3** fast_sample$(\boldsymbol{\mu}, \mathbf{U}, \mathbf{d})$ |
| 7: $\quad \mathbf{V} \leftarrow$ fast_eig$(\delta \mathbf{u}_1, .., \delta \mathbf{u}_L, \beta \mathbf{g}_1, .., \beta \mathbf{g}_M, L)$ | 1: $\boldsymbol{\epsilon} \sim \mathcal{N}(0, \mathbf{I}_D)$ |
| 8: $\quad \Delta_d \leftarrow \sum_{i=1}^{L} \delta \mathbf{u}_i^2 + \sum_{i=1}^{M} \beta \mathbf{g}_i^2 - \sum_{i=1}^{L} \mathbf{v}_i^2$ | 2: $\mathbf{V} \leftarrow \mathbf{d}^{-1/2} \odot \mathbf{U}$ |
| 9: $\quad \mathbf{U} \leftarrow \mathbf{V}$ | 3: $\mathbf{A} \leftarrow$ Cholesky$(\mathbf{V}^\top \mathbf{V})$ |
| 10: $\quad \mathbf{d} \leftarrow \delta \mathbf{d} + \Delta_d + \lambda \mathbf{1}$ | 4: $\mathbf{B} \leftarrow$ Cholesky$(\mathbf{I}_L + \mathbf{V}^\top \mathbf{V})$ |
| 11: $\quad \hat{\mathbf{g}} \leftarrow \sum_i \mathbf{g}_i + \lambda \boldsymbol{\mu}$ | 5: $\mathbf{C} \leftarrow \mathbf{A}^{-\top}(\mathbf{B} - \mathbf{I}_L)\mathbf{A}^{-1}$ |
| 12: $\quad \Delta_\mu \leftarrow$ fast_inverse$(\hat{\mathbf{g}}, \mathbf{U}, \mathbf{d})$ | 6: $\mathbf{K} \leftarrow (\mathbf{C} + \mathbf{V}^\top \mathbf{V})^{-1}$ |
| 13: $\quad \boldsymbol{\mu} \leftarrow \boldsymbol{\mu} - \alpha \Delta_\mu$ | 7: $\mathbf{y} \leftarrow \mathbf{d}^{-1/2}\boldsymbol{\epsilon} - \mathbf{V}\mathbf{K}\mathbf{V}^\top \boldsymbol{\epsilon}$ |
| 14: **end while** | 8: **return** $\boldsymbol{\mu} + \mathbf{y}$ |
| 15: **return** $\boldsymbol{\mu}, \mathbf{U}, \mathbf{d}$ | |

Figure 3: Algorithm 1 gives the pseudo-code for SLANG. Here, $M$ is the minibatch size, $L$ is the number of low-rank factors, $\lambda$ is the prior precision parameter, and $\alpha, \beta$ are learning rates. The diagonal component is denoted with a vector $\mathbf{d}$ and the columns of the matrix $\mathbf{U}$ and $\mathbf{V}$ are denoted by $\mathbf{u}_j$ and $\mathbf{v}_j$ respectively. The algorithm depends on multiple subroutines, described in more details in Section 3.1. The overall complexity of the algorithm is $O(DL^2 + DLM)$.

or 10), this only adds a small multiplicative constant in the runtime. SLANG reduces the cost of the update (2) significantly while preserving some posterior correlations.

## 4 Experiments

In this section, our goal is to show experimental results in support of the following claims: (1) SLANG gives reasonable posterior approximations, and (2) SLANG performs well on standard benchmarks for Bayesian neural networks. We present evaluations on several LIBSVM datasets, the UCI regression benchmark, and MNIST classification. SLANG beats mean-field methods on almost all tasks considered and performs comparably to state-of-the-art methods. SLANG also converges faster than mean-field methods.

### 4.1 Bayesian Logistic Regression

We considered four benchmark datasets for our comparison: USPS 3vs5, Australian, Breast-Cancer, and a1a. Details of the datasets are in Table 8 in Appendix E.2 along with the implementation details of the methods we compare to. We use L-BFGS [37] to compute the optimal full-Gaussian variational approximation that minimizes the ELBO using the method described in Marlin et al. [22]. We refer to the optimal full-Gaussian variational approximation as the "Full-Gaussian Exact" method. We also compute the optimal mean-field Gaussian approximation and refer to it as "MF Exact".

Figure 1 shows a qualitative comparison of the estimated posterior means, variances, and covariances for the USPS-3vs5 dataset ($N = 770, D = 256$). The figure on the left compares covariance approximations obtained with SLANG to the Full-Gaussian Exact method. Only off-diagonal entries are shown. We see that the approximation becomes more and more accurate as the rank is increased. The figures on the right compare the means and variances. The means match closely for all methods, but the variance is heavily underestimated by the MF Exact method; we see that the variances obtained under the mean-field approximation estimate a high variance where Full-Gaussian Exact has a low variance and vice-versa. This "trend-reversal" is due to the typical shrinking behavior of mean-field methods [36]. In contrast, SLANG corrects the trend reversal problem even when $L = 1$. Similar results for other datasets are shown in Figure 7 in Appendix E.1.

Table 1: Results on Bayesian logistic regression where we compare SLANG to three full-Gaussian methods and three mean-field methods. We measure negative ELBO, test log-loss, and symmetric KL-divergence between each approximation and the Full-Gaussian Exact method (last column). Lower values are better. SLANG nearly always gives better results than the mean-field methods, and with $L = 10$ is comparable to Full-Gaussian methods. This shows that our structured covariance approximation is reasonably accurate for Bayesian logistic regression.

| Dataset | Metrics | Mean-Field Methods | | | SLANG | | | Full Gaussian | | |
|---|---|---|---|---|---|---|---|---|---|---|
| | | **EF** | **Hess.** | **Exact** | **L = 1** | **L = 5** | **L = 10** | **EF** | **Hess.** | **Exact** |
| Australian | ELBO | 0.614 | 0.613 | 0.593 | 0.574 | 0.569 | **0.566** | 0.560 | 0.558 | 0.559 |
| | NLL | 0.348 | 0.347 | 0.341 | 0.342 | 0.339 | **0.338** | 0.340 | 0.339 | 0.338 |
| | KL ( $\times 10^4$) | 2.240 | 2.030 | 0.195 | 0.033 | 0.008 | **0.002** | 0.000 | 0.000 | 0.000 |
| Breast Cancer | ELBO | 0.122 | 0.121 | 0.121 | 0.112 | 0.111 | **0.111** | 0.111 | 0.109 | 0.109 |
| | NLL | 0.095 | 0.094 | 0.094 | 0.092 | 0.092 | **0.092** | 0.092 | 0.091 | 0.091 |
| | KL ( $\times 10^0$) | 8.019 | 9.071 | 7.771 | 0.911 | 0.842 | **0.638** | 0.637 | 0.002 | 0.000 |
| a1a | ELBO | 0.384 | 0.383 | 0.383 | 0.377 | 0.374 | **0.373** | 0.369 | 0.368 | 0.368 |
| | NLL | 0.339 | 0.339 | 0.339 | 0.339 | 0.339 | **0.339** | 0.339 | 0.339 | 0.339 |
| | KL ($\times 10^2$) | 2.590 | 2.208 | 1.295 | 0.305 | 0.173 | **0.118** | 0.014 | 0.000 | 0.000 |
| USPS 3vs5 | ELBO | 0.268 | 0.268 | 0.267 | 0.210 | 0.198 | **0.193** | 0.189 | 0.186 | 0.186 |
| | NLL | 0.139 | 0.139 | 0.138 | 0.132 | 0.132 | **0.131** | 0.131 | 0.130 | 0.130 |
| | KL ($\times 10^1$) | 7.684 | 7.188 | 7.083 | 1.492 | 0.755 | **0.448** | 0.180 | 0.001 | 0.000 |

The complete results for Bayesian logistic regression are summarized in Table 1, where we also compare to four additional methods called "Full-Gaussian EF", "Full-Gaussian Hessian", "Mean-Field EF", and "Mean-Field Hessian". The Full-Gaussian EF method is the natural-gradient update (2) which uses the EF matrix $\hat{\mathbf{G}}(\boldsymbol{\theta})$, while the Full-Gaussian Hessian method uses the Hessian instead of the EF matrix (the updates are given in (12) and (13) in Appendix A). The last two methods are the mean-field versions of the Full-Gaussian EF and Full-Gaussian Hessian methods, respectively. We compare negative ELBO, test log-loss using cross-entropy, and symmetric KL-divergence between the approximations and the Full-Gaussian Exact method. We report averages over 20 random 50%-50% training-test splits of the dataset. Variances and results for SLANG with $L = 2$ are omitted here due to space constraints, but are reported in Table 6 in Appendix E.1.

We find that SLANG with $L = 1$ nearly always produces better approximations than the mean-field methods. As expected, increasing $L$ improves the quality of the variational distribution found by SLANG according to all three metrics. We also note that Full-Gaussian EF method has similar performance to the Full-Gaussian Hessian method, which indicates that the EF approximation may be acceptable for Bayesian logistic regression.

The left side in Figure 4 shows convergence results on the USPS 3vs5 and Breast Cancer datasets. The three methods SLANG(1, 2, 3) refer to SLANG with $L = 1, 5, 10$. We compare these three SLANG methods to Mean-Field Hessian and Full-Gaussian Hessian. SLANG converges faster than the mean-field method, and matches the convergence of the full-Gaussian method when $L$ is increased.

## 4.2 Bayesian Neural Networks (BNNs)

An example for Bayesian Neural Networks on a synthetic regression dataset is given in Appendix F.1, where we illustrate the quality of SLANG's posterior covariance.

The right side in Figure 4 shows convergence results for the USPS 3vs5 and Breast Cancer datasets. Here, the three methods SLANG(1, 2, 3) refer to SLANG with $L = 8, 16, 32$. We compare SLANG to a mean-field method called Bayes by Backprop [7]. Similar to the Bayesian logistic regression experiment, SLANG converges much faster than the mean-field method. However, the ELBO convergence for SLANG shows that the optimization procedure does not necessarily converge to a local minimum. This issue does not appear to affect the test log-likelihood. While it might only be due to stochasticity, it is possible that the problem is exacerbated by the replacement of the Hessian with the EF matrix. We have not determined the specific cause and it warrants further investigation in future work.

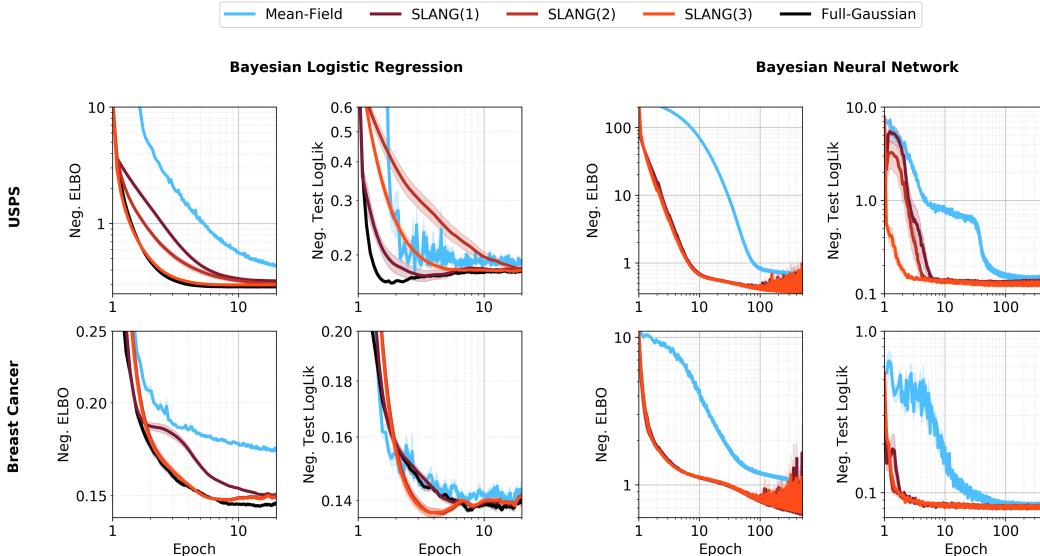

Figure 4: This figure compares the convergence behavior on two datasets: USPS 3vs5 (top) and Breast Cancer (bottom); and two models: Bayesian logistic regression (left) and Bayesian neural networks (BNN) (right). The three methods SLANG(1, 2, 3) refer to SLANG with $L = 1, 5, 10$ for logistic regression. For BNN, they refer to SLANG with $L = 8, 16, 32$. The mean-field method is a natural-gradient mean-field method for logistic regression (see text) and BBB [7] for BNN. This comparison clearly shows that SLANG converges faster than the mean-field method, and, for Bayesian logistic regression, matches the convergence of the full-Gaussian method when $L$ is increased.

Table 2: Comparison on UCI datasets using Bayesian neural networks. We repeat the setup used in Gal and Ghahramani [10]. SLANG uses $L = 1$, and outperforms BBB but gives comparable performance to Dropout.

| Dataset | Test RMSE | | | Test log-likelihood | | |
|---|---|---|---|---|---|---|
| | **BBB** | **Dropout** | **SLANG** | **BBB** | **Dropout** | **SLANG** |
| Boston | $3.43 \pm 0.20$ | $\mathbf{2.97 \pm 0.19}$ | $3.21 \pm 0.19$ | $-2.66 \pm 0.06$ | $\mathbf{-2.46 \pm 0.06}$ | $-2.58 \pm 0.05$ |
| Concrete | $6.16 \pm 0.13$ | $\mathbf{5.23 \pm 0.12}$ | $5.58 \pm 0.19$ | $-3.25 \pm 0.02$ | $\mathbf{-3.04 \pm 0.02}$ | $-3.13 \pm 0.03$ |
| Energy | $0.97 \pm 0.09$ | $1.66 \pm 0.04$ | $\mathbf{0.64 \pm 0.03}$ | $-1.45 \pm 0.10$ | $-1.99 \pm 0.02$ | $\mathbf{-1.12 \pm 0.01}$ |
| Kin8nm | $0.08 \pm 0.00$ | $0.10 \pm 0.00$ | $\mathbf{0.08 \pm 0.00}$ | $\mathbf{1.07 \pm 0.00}$ | $0.95 \pm 0.01$ | $1.06 \pm 0.00$ |
| Naval | $0.00 \pm 0.00$ | $0.01 \pm 0.00$ | $\mathbf{0.00 \pm 0.00}$ | $4.61 \pm 0.01$ | $3.80 \pm 0.01$ | $\mathbf{4.76 \pm 0.00}$ |
| Power | $4.21 \pm 0.03$ | $\mathbf{4.02 \pm 0.04}$ | $4.16 \pm 0.04$ | $-2.86 \pm 0.01$ | $\mathbf{-2.80 \pm 0.01}$ | $-2.84 \pm 0.01$ |
| Wine | $0.64 \pm 0.01$ | $\mathbf{0.62 \pm 0.01}$ | $0.65 \pm 0.01$ | $-0.97 \pm 0.01$ | $\mathbf{-0.93 \pm 0.01}$ | $-0.97 \pm 0.01$ |
| Yacht | $1.13 \pm 0.06$ | $1.11 \pm 0.09$ | $\mathbf{1.08 \pm 0.06}$ | $-1.56 \pm 0.02$ | $\mathbf{-1.55 \pm 0.03}$ | $-1.88 \pm 0.01$ |

Next, we present results on the UCI regression datasets which are common benchmarks for Bayesian neural networks [14]. We repeat the setup[2] used in Gal and Ghahramani [10]. Following their work, we use neural networks with one hidden layer with 50 hidden units and ReLU activation functions. We compare SLANG with $L = 1$ to the Bayes By Backprop (BBB) method [7] and the Bayesian Dropout method of [10]. For the 5 smallest datasets, we used a mini-batch size of 10 and 4 Monte-Carlo samples during training. For the 3 larger datasets, we used a mini-batch size of 100 and 2 Monte-Carlo samples during training. More details are given in Appendix F.3. We report test RMSE and test log-likelihood in Table 2. SLANG with just one rank outperforms BBB on 7 out of 8 datasets for RMSE and on 5 out of 8 datasets for log-likelihood. Moreover, SLANG shows comparable performance to Dropout.

Table 3: Comparison of SLANG on the MNIST dataset. We use a two layer neural network with 400 units each. SLANG obtains good performances for moderate values of $L$.

|  | BBB | SLANG | | | | | |
| --- | --- | --- | --- | --- | --- | --- | --- |
|  |  | L = 1 | L = 2 | L = 4 | L = 8 | L = 16 | L = 32 |
| Test Error | 1.82% | 2.00% | 1.95% | 1.81% | 1.92% | 1.77% | **1.73%** |

Finally, we report results for classification on MNIST. We train a BNN with two hidden layers of 400 hidden units each. The training set consists of 50,000 examples and the remaining 10,000 are used as a validation set. The test set is a separate set which consists of 10,000 examples. We use SLANG with $L = 1, 2, 4, 8, 16, 32$. For each value of $L$, we choose the prior precision and learning rate based on performance on the validation set. Further details can be found in Appendix F.4. The test accuracies are reported in Table 3 and compared to the results obtained in [7] by using BBB. For SLANG, a good performance can be obtained for a moderate $L$. Note that there might be small differences between our experimental setup and the one used in [7] since BBB implementation is not publicly available. Therefore, the results might not be directly comparable. Nevertheless, SLANG appears to perform well compared to BBB.

## 5 Conclusion

We consider the challenging problem of uncertainty estimation in large deep models. For such problems, it is infeasible to form a Gaussian approximation to the posterior distribution. We address this issue by estimating a Gaussian approximation that uses a covariance with low-rank plus diagonal structure. We proposed an approximate natural-gradient algorithm to estimate the structured covariance matrix. Our method, called SLANG, relies only on the back-propagated gradients to estimate the covariance structure, which is a desirable feature when working with deep models. Empirical results strongly suggest that the accuracy of our method is better than those obtained by using mean-field methods. Moreover, we observe that, unlike mean-field methods, our method does not drastically shrink the marginal variances. Experiments also show that SLANG is highly flexible and that its accuracy can be improved by increasing the rank of the covariance's low-rank component. Finally, our method converges faster than the mean-field methods and can sometimes converge as fast as VI methods that use a full-Gaussian approximation.

The experiments presented in this paper are restricted to feed-forward neural networks. This is partly because existing deep-learning software packages do not support individual gradient computations. Individual gradients, which are required in line 6 of Algorithm 1, must be manually implemented for other types of architectures. Further work is therefore necessary to establish the usefulness of our method on other types of network architectures.

SLANG is based on a natural-gradient method that employs the empirical Fisher approximation [19]. Our empirical results suggest that this approximation is reasonably accurate. However, it is not clear if this is always the case. It is important to investigate this issue to gain better understanding of the effect of the approximation, both theoretically and empirically.

During this work, we also found that comparing the quality of covariance approximations is a nontrivial task for deep neural networks. We believe that existing benchmarks are not sufficient to establish the quality of an approximate Bayesian inference method for deep models. An interesting and useful area of further research is the development of good benchmarks that better reflect the quality of posterior approximations. This will facilitate the design of better inference algorithms.

**Acknowledgements**

We thank the anonymous reviewers for their helpful feedback. We greatly appreciate useful discussions with Shun-ichi Amari (RIKEN), Rio Yokota (Tokyo Institute of Technology), Kazuki Oosawa (Tokyo Institute of Technology), Wu Lin (University of British Columbia), and Voot Tangkaratt (RIKEN). We are also thankful for the RAIDEN computing system and its support team at the RIKEN Center for Advanced Intelligence Project, which we used extensively for our experiments.

## Footnotes

[2]We use the data splits available at `https://github.com/yaringal/DropoutUncertaintyExps`

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
