[Supplementary Material]

# A Derivation of the VOGN Update

We can derive the VOGN update by using the Variational Online Newton (VON) method derived in Appendix D of [19]. The VON updates are given as follows:

$$\boldsymbol{\mu}_{t+1} = \boldsymbol{\mu}_t - \beta_t \ \boldsymbol{\Sigma}_{t+1} \left[ \hat{\mathbf{g}}(\boldsymbol{\theta}_t) + \lambda \boldsymbol{\mu}_t \right], \tag{12}$$

$$\boldsymbol{\Sigma}_{t+1}^{-1} = (1 - \beta_t)\boldsymbol{\Sigma}_t^{-1} + \ \beta_t \ \left[ \hat{\mathbf{H}}(\boldsymbol{\theta}_t) + \lambda \mathbf{I} \right], \tag{13}$$

where $t$ is the iteration number, $\beta_t > 0$ is the learning rate, $\boldsymbol{\theta}_t \sim \mathcal{N}(\boldsymbol{\theta}|\boldsymbol{\mu}_t, \boldsymbol{\Sigma}_t)$, and $\hat{\mathbf{g}}(\boldsymbol{\theta}_t)$ and $\hat{\mathbf{H}}(\boldsymbol{\theta}_t)$ are the stochastic gradient and Hessian, defined respectively as follows:

$$\hat{\mathbf{g}}(\boldsymbol{\theta}_t) := -\frac{N}{M} \sum_{i \in \mathcal{M}} \mathbf{g}_i(\boldsymbol{\theta}_t), \tag{14}$$

$$\hat{\mathbf{H}}(\boldsymbol{\theta}_t) := -\frac{N}{M} \sum_{i \in \mathcal{M}} \nabla_{\theta\theta}^2 \log p(\mathcal{D}_i|\boldsymbol{\theta}_t). \tag{15}$$

$\mathbf{g}_i(\boldsymbol{\theta}_t) := \nabla_\theta \log p(\mathcal{D}_i|\boldsymbol{\theta}_t)$ is the back-propagated gradient obtained on the $i$'th data example, and $\mathcal{M}$ is a minibatch of $M$ data examples.

Dealing with Hessians can be difficult because they may not always be positive-definite and may produce invalid Gaussian approximations. Following [19], we approximate the Hessian by the Empirical Fisher (EF) matrix:

$$\text{EF:} \quad \hat{\mathbf{H}}(\boldsymbol{\theta}) \approx \hat{\mathbf{G}}(\boldsymbol{\theta}) := \frac{N}{M} \sum_{i \in \mathcal{M}} \mathbf{g}_i(\boldsymbol{\theta})\mathbf{g}_i(\boldsymbol{\theta})^\top. \tag{16}$$

This is also known as the Generalized Gauss-Newton approximation. Using this approximation in (13) gives us the VOGN update of (2).

The VON update is an exact natural-gradient method and uses a single learning rate $\beta$. The VOGN update, on the other hand, is an approximate natural-gradient method because it uses the EF approximation. Due to this approximation, a single learning rate might not give good results and it can be sensible to use different learning rates for the $\boldsymbol{\mu}$ and $\boldsymbol{\Sigma}$ updates. In (2), we therefore use a different learning rate for $\boldsymbol{\mu}$ (denoted by $\alpha$).

# B An Alternative Low-Rank Update

We tried an alternative approach to learn the low-rank plus diagonal covariance approximation. We call this method SLANG-OnlineEig. It forms the low-rank term in the precision approximation from an online estimate of the $L$ leading eigenvectors of $\hat{\mathbf{G}}(\boldsymbol{\theta})$. We now describe this procedure in detail. Experimental results are presented and comparisons are made with SLANG.

## B.1 Approximating Natural Gradients by Online Estimation of the Eigendecomposition (SLANG-OnlineEig)

The following eigenvalue decomposition forms the basis of SLANG-OnlineEig:

$$\hat{\mathbf{G}}(\boldsymbol{\theta}) \approx \mathbf{Q}_{1:L}\boldsymbol{\Lambda}_{1:L}\mathbf{Q}_{1:L}^\top = \mathbf{Q}_{1:L}\boldsymbol{\Lambda}_{1:L}^{1/2}(\mathbf{Q}_{1:L}\boldsymbol{\Lambda}_{1:L}^{1/2})^\top. \tag{17}$$

We emphasize that SLANG-OnlineEig involves the decomposition of $N\hat{\mathbf{G}}(\boldsymbol{\theta})$, rather than the updated matrix $(1 - \beta)\mathbf{U}_t\mathbf{U}_t^\top + \beta N\hat{\mathbf{G}}(\boldsymbol{\theta})$ as in SLANG. This is cheaper by a factor of $O(DL^2)$, which is a marginal difference as $L \ll D$ in most applications. To mimic the update of $\boldsymbol{\Sigma}^{-1}$ in (2), we use the following "moving-average" update for $\mathbf{U}$:

$$\mathbf{U}_{t+1} = (1 - \beta_t)\mathbf{U}_t + \beta_t\mathbf{Q}_{1:L}\boldsymbol{\Lambda}_{1:L}^{1/2}, \tag{18}$$

where $\beta_t \in [0, 1]$ is a scalar learning rate. $\mathbf{U}_{t+1}$ is a thus an online estimate of weighted eigenvectors of the EF.

Similar to SLANG, the diagonal $\mathbf{D}$ is updated to capture the curvature information lost in the projection of $\hat{\mathbf{G}}(\boldsymbol{\theta})$ to $\mathbf{Q}_{1:L}\boldsymbol{\Lambda}_{1:L}\mathbf{Q}_{1:L}^{\top}$, i.e., the remaining $M - L$ eigenvectors:

$$\mathbf{D}_{t+1} = (1 - \delta_t)\mathbf{D}_t + \delta_t \left[ \text{diag}(\hat{\mathbf{G}}(\boldsymbol{\theta}_t)) - \text{diag}(\mathbf{Q}_{1:L}\boldsymbol{\Lambda}_{1:L}\mathbf{Q}_{1:L}^{\top} + \lambda \mathbf{I}) \right]. \tag{19}$$

The updated covariance is then given by

$$\hat{\boldsymbol{\Sigma}}_{t+1}^{-1} := \mathbf{U}_{t+1}\mathbf{U}_{t+1}^{\top} + \mathbf{D}_{t+1}. \tag{20}$$

The final step of SLANG-OnlineEig is identical to equation (11):

$$\text{SLANG-Online-Eig:} \quad \boldsymbol{\mu}_{t+1} = \boldsymbol{\mu}_t - \alpha_t \hat{\boldsymbol{\Sigma}}_{t+1} \left[ \hat{\mathbf{g}}(\boldsymbol{\theta}_t) + \lambda\boldsymbol{\mu}_t \right]. \tag{21}$$

SLANG-OnlineEig is amenable to the same algorithmic tools as SLANG, which were discussed in 3.1. Goodfellow's trick can be used to compute the Jacobian needed for the EF and algorithms 2 and 3 are available for fast covariance-vector products and fast sampling, respectively. The overall computational complexity is $O(DL^2 + DLM)$ and memory cost is $O(DL + DM)$.

## B.2 Comparison with SLANG

SLANG-OnlineEig has several promising properties. It has slightly better computational complexity than SLANG and its update closely resembles the natural gradient update in (2). However, update (18) involves the product approximation,

$$\mathbf{U}_{t+1}\mathbf{U}_{t+1}^{\top} = \left( (1-\beta_t)\mathbf{U}_t + \beta_t\mathbf{Q}_{1:L}\boldsymbol{\Lambda}_{1:L}^{1/2} \right)\left( (1-\beta_t)\mathbf{U}_t + \beta_t\mathbf{Q}_{1:L}\boldsymbol{\Lambda}_{1:L}^{1/2} \right)^{\top}$$

$$\approx (1-\beta_t)\mathbf{U}_t\mathbf{U}_t^{\top} + \beta_t\mathbf{Q}_{1:L}\boldsymbol{\Lambda}_{1:L}^{1/2}\left( \mathbf{Q}_{1:L}\boldsymbol{\Lambda}_{1:L}^{1/2} \right)^{\top},$$

where the second line is the true product of interest. The update assumes that the online estimate of the factors of the eigendecomposition well-approximates the online estimate of the eigendecomposition itself.

SLANG does not require the product approximation for efficient covariance learning. Instead, $\mathbf{U}_t\mathbf{U}_t^{\top}$ is updated exactly before the projection into the space of rank-$L$ matrices. This is why SLANG with $L = D$ reduces to 2 using the EF approximation. SLANG-OnlineEig does not posses this property because of the product approximation. It also does not necessarily match precision diagonals with the $L = D$ update, as SLANG does for all $L < D$.

A final issue is that SLANG-OnlineEig also requires matching stochastic eigenvector estimates to their corresponding online estimates in $\mathbf{U}_{t+1}$. This may introduce additional approximation error when $\hat{\mathbf{G}}(\boldsymbol{\theta})$ is highly stochastic.

## B.3 Experimental Results for SLANG-OnlineEig

Table 4 compares SLANG and SLANG-OnlineEig for logistic regression on the Australian, Breast Cancer, USPS 3vs5, and a1a datasets from LIBSVM. The results presented for SLANG are identical to those in Table 1. SLANG always matches or beats the best results for SLANG-OnlineEig. Furthermore, as $L$ is increased, the quality of posterior approximations computed by SLANG improves while SLANG-OnlineEig approximations sometimes degrade. We speculate that this is due to the product approximation.

Figure 5 presents results on the convergence of SLANG-OnlineEig for logistic regression and regression with a Bayesian neural network.

Table 5 shows regression on the UCI datasets using Bayesian neural networks. The setup for this experiment was similar to the experiment in Section 4.2, except that the learning rates were fixed to $\alpha = 0.01$ and $\beta = (0.9, 0.999)$ for all datasets, both for SLANG-OnlineEig and for the Adam optimizer used for BBB. Moreover, the search spaces for the Bayesian optimization were fixed (using a normalized scale for the noise precision) and not adjusted to the individual datasets. Finally, BBB used 40 MC samples for the 5 smallest datasets and 20 MC samples for the 3 largest datasets in this experiment.

Table 4: Comparison of SLANG and SLANG-OnlineEig for logistic regression. SLANG obtains as good or better loss under every metric for each dataset. Additionally, the quality of posterior approximations computed by SLANG improves while SLANG-OnlineEig approximations sometimes degrade as $L$ is increased. The best result for each method is in bold.

| | | SLANG-OnlineEig | | | SLANG | | |
|---|---|---|---|---|---|---|---|
| | Metrics | L = 1 | L = 2 | L = 5 | L = 1 | L = 2 | L = 5 |
| Australian | ELBO | 0.580 | **0.578** | 0.652 | 0.574 | 0.574 | **0.569** |
| | NLL | 0.344 | **0.343** | 0.347 | 0.342 | 0.342 | **0.339** |
| | KL ($\times 10^4$) | 0.057 | 0.032 | **0.012** | 0.033 | 0.031 | **0.008** |
| Breast Cancer | ELBO | **0.114** | 0.116 | 0.123 | 0.112 | 0.111 | **0.111** |
| | NLL | **0.092** | 0.093 | 0.093 | 0.092 | 0.092 | **0.092** |
| | KL ($\times 10^0$) | **1.544** | 2.128 | 4.402 | 0.911 | 0.756 | **0.842** |
| a1a | ELBO | 0.380 | **0.380** | 0.383 | 0.377 | 0.376 | **0.374** |
| | NLL | 0.339 | 0.339 | **0.339** | 0.339 | 0.339 | **0.339** |
| | KL ($\times 10^2$) | 0.351 | 0.293 | **0.253** | 0.305 | 0.249 | **0.173** |
| USPS 3vs5 | ELBO | 0.210 | **0.208** | 0.210 | 0.210 | 0.206 | **0.198** |
| | NLL | 0.133 | **0.132** | 0.133 | 0.132 | 0.132 | **0.132** |
| | KL ($\times 10^1$) | 1.497 | **1.353** | 1.432 | 1.492 | 1.246 | **0.755** |

Figure 5: (Left) This figure shows the convergence behavior of SLANG-OnlineEig on logistic regression for USPS. We plot the KL divergence between Full-Gaussian and the approximate posterior. The higher rank approximations give better results and all of them beat mean-field. (Middle) This figure summarizes the results of 5 runs, showing that as we increase $L$ the approximation gets better. (Right) We show convergence of a Bayesian neural network on the Energy dataset. We see that better structured approximation leads to faster convergence.

## C  Additional Algorithmic Details for SLANG

The following section gives more detail about the individual components of the algorithm required to leverage the low-rank plus diagonal structure for computational efficiency. In general, we obtain efficient algorithms by operating only on $D \times L$ or $L \times L$ matrices. This avoids the $O(D^2)$ storage cost and the $O(D^3)$ computational cost of working in the $D \times D$ space.

### C.1  Fast Computation of Individual Gradients

Most deep-learning automatic differentiation packages, such as PyTorch [27] and TensorFlow [1], are optimized to return the *overall* gradient of a minibatch, not *individual* gradients for each example passed through the network. It is true that the naïve option of doing a forward and backward pass for each example has a similar computational complexity as a fully parallel version. However, in practice

Table 5: Predictive performance on UCI datasets using Bayesian neural networks where SLANG-OnlineEig beats BBB and performs comparably to Dropout.

| | Test RMSE | | | Test log-likelihood | | |
|---|---|---|---|---|---|---|
| **Dataset** | **BBB** | **Dropout** | **OnlineEig** | **BBB** | **Dropout** | **OnlineEig** |
| Boston | $4.04 \pm 0.28$ | $\mathbf{2.97 \pm 0.19}$ | $3.17 \pm 0.17$ | $-2.75 \pm 0.07$ | $\mathbf{-2.46 \pm 0.06}$ | $-2.61 \pm 0.06$ |
| Concrete | $6.16 \pm 0.14$ | $\mathbf{5.23 \pm 0.12}$ | $5.79 \pm 0.13$ | $-3.22 \pm 0.02$ | $\mathbf{-3.04 \pm 0.02}$ | $-3.19 \pm 0.02$ |
| Energy | $0.86 \pm 0.04$ | $1.66 \pm 0.04$ | $\mathbf{0.59 \pm 0.01}$ | $-1.20 \pm 0.05$ | $-1.99 \pm 0.02$ | $\mathbf{-1.05 \pm 0.01}$ |
| Kin8nm | $0.09 \pm 0.00$ | $0.10 \pm 0.00$ | $\mathbf{0.08 \pm 0.00}$ | $0.97 \pm 0.01$ | $0.95 \pm 0.01$ | $\mathbf{1.13 \pm 0.00}$ |
| Naval | $\mathbf{0.00 \pm 0.00}$ | $0.01 \pm 0.00$ | $\mathbf{0.00 \pm 0.00}$ | $\mathbf{5.34 \pm 0.07}$ | $3.80 \pm 0.01$ | $5.09 \pm 0.08$ |
| Power | $4.28 \pm 0.03$ | $\mathbf{4.02 \pm 0.04}$ | $4.09 \pm 0.04$ | $-2.87 \pm 0.01$ | $\mathbf{-2.80 \pm 0.01}$ | $-2.83 \pm 0.01$ |
| Wine | $0.66 \pm 0.01$ | $\mathbf{0.62 \pm 0.01}$ | $0.64 \pm 0.01$ | $-0.99 \pm 0.01$ | $\mathbf{-0.93 \pm 0.01}$ | $-0.98 \pm 0.01$ |
| Yacht | $1.07 \pm 0.08$ | $1.11 \pm 0.09$ | $\mathbf{0.72 \pm 0.04}$ | $-1.45 \pm 0.03$ | $-1.55 \pm 0.03$ | $\mathbf{-1.41 \pm 0.01}$ |

most of the computations involved can be either reused across examples, or sped up drastically by batching them. Implementations that use matrix-matrix multiplications instead of repeatedly doing matrix-vector operations are far more efficient on GPUs.

Ian Goodfellow's note [11] outlines a method for efficiently computing per-example gradients. It suggests saving the neuron activations and the linear combinations of activations computed during the minibatch's forward pass through the neural network. These values are then used in a manual implementation of the per-example gradients, which avoids the summation defined by the cost function. While much more efficient than the sequential approach, Goodfellow's approach requires more implementation effort; a separate implementation is required to handle each type of layer used. This is partly why the experiments presented here are limited to standard Multi-Layer Perceptrons. We hope to improve upon this in future implementations.

## C.2 Fast Top-$L$ Eigendecomposition

The goal is to get the top-$L$ eigenvalues and eigenvectors of the matrix $(1 - \beta)\mathbf{U}_t\mathbf{U}_t^\top + \beta\hat{\mathbf{G}}(\boldsymbol{\theta}_t)$ defined in (6). Since we do not want to compute the $D \times D$ matrix explicitly, we use the low-rank structure of the update matrix to compute matrix-vector or matrix-matrix products with computations in $O(DL + DM)$. This can be seen by rewriting the matrix as follows:

$$(1 - \beta)\mathbf{U}_t\mathbf{U}_t^\top + \beta\hat{\mathbf{G}}(\boldsymbol{\theta}_t) = (1 - \beta)\sum_{l=1}^{L} \mathbf{u}_t^{(l)}\mathbf{u}_t^{(l)\top} + \beta\frac{N}{M}\sum_{i\in\mathcal{M}} \mathbf{g}_i(\boldsymbol{\theta}_t)\mathbf{g}_i(\boldsymbol{\theta}_t)^\top. \quad (22)$$

These products can be used to compute eigendecomposition efficiently by using a randomized algorithm.

The main idea behind the randomized eigendecomposition is to project a matrix $\mathbf{A}$ onto a randomly selected subspace by sampling $K$ vectors $\boldsymbol{\epsilon}_k \in \mathbb{R}^D$, each entry being selected uniformly at random, and computing $\mathbf{A}_K = \mathbf{A}[\boldsymbol{\epsilon}_1, ..., \boldsymbol{\epsilon}_K]$, where $K$ is larger than $L$. A traditional eigendecomposition can then be performed on the $D \times K$ matrix $\mathbf{A}_K$ to recover the top $L$ eigenvectors, with $K$ acting as a precision-computation tradeoff parameter. More details on randomized eigenvalue methods can be found in [13].

Our implementation of this procedure follows Facebook's Fast Randomized SVD[3] closely; starting with a random matrix, multiplies it by $\mathbf{A}$ and applies a QR decomposition on the result. This process is repeated on the resulting matrix for a few iterations to improve stability, similarly to the Lanczos iterations. As all operations are done on the smaller $D \times K$ matrix, using $K = L + 2$ as recommended in [3]), the computational cost of the QR decomposition and eigendecomposition are in $O(DL^2)$, leading to an $O(DL^2 + DM)$ algorithm overall.

## C.3 Fast multiplication by inverse of low-rank + diagonal

To implement the natural gradient update (11), we need to be able to multiply an arbitrary vector by $\hat{\mathbf{\Sigma}}$, given $\hat{\mathbf{\Sigma}}^{-1} = \mathbf{U}\mathbf{U}^\top + \mathbf{D}$. Woodbury's identity can be used to do so without forming the $D \times D$ matrix and doing the costly $O(D^3)$ inversion. The identity gives

$$(\mathbf{D} + \mathbf{U}\mathbf{U}^\top)^{-1} = \mathbf{D}^{-1} - \mathbf{D}^{-1}\mathbf{U}(\mathbf{I}_L + \mathbf{U}^\top\mathbf{D}^{-1}\mathbf{U})^{-1}\mathbf{U}^\top\mathbf{D}^{-1}. \tag{23}$$

The only inversions remaining involve diagonal or $L \times L$ matrices. Correct ordering of the operations allows the $O(D^2)$ storage cost to be avoided when computing the product $(\mathbf{U}\mathbf{U}^\top + \mathbf{D})^{-1}\mathbf{x}$ and ensures that we only need to store $D \times L, L \times L$ or diagonal matrices,

$$\left(\mathbf{D} + \mathbf{U}\mathbf{U}^\top\right)^{-1}\mathbf{x} = (\mathbf{D}^{-1}\mathbf{x}) - \mathbf{D}^{-1}\left(\mathbf{U}\left(\left(\mathbf{I}_L + \mathbf{U}^\top\mathbf{D}^{-1}\mathbf{U}\right)^{-1}\left(\mathbf{U}^\top\left(\mathbf{D}^{-1}\mathbf{x}\right)\right)\right)\right), \tag{24}$$

yielding a $O(DL^2)$ algorithm.

## C.4 Fast sampling

To generate a sample from $\mathcal{N}(\boldsymbol{\mu}, \hat{\mathbf{\Sigma}})$, it is sufficient to generate a sample $\boldsymbol{\epsilon} \sim \mathcal{N}(0, \mathbf{I}_D)$ and compute $\boldsymbol{\mu} + \mathbf{A}\boldsymbol{\epsilon}$, where $\mathbf{A}\mathbf{A}^\top = \hat{\mathbf{\Sigma}}$. $\hat{\mathbf{\Sigma}}$ can be factorized efficiently by exploiting its "low-rank plus diagonal" structure:

$$\hat{\mathbf{\Sigma}} = \left(\mathbf{U}\mathbf{U}^\top + \mathbf{D}\right)^{-1}, \tag{25}$$

$$= \left(\mathbf{D}^{1/2}\left(\underbrace{\mathbf{D}^{-1/2}\mathbf{U}\mathbf{U}^\top\mathbf{D}^{-1/2}}_{\mathbf{V}\mathbf{V}^\top} + \mathbf{I}\right)\mathbf{D}^{1/2}\right)^{-1}, \tag{26}$$

$$= \mathbf{D}^{-1/2}\left(\mathbf{V}\mathbf{V}^\top + \mathbf{I}\right)^{-1}\mathbf{D}^{-1/2}. \tag{27}$$

Letting $\mathbf{W}$ be a symmetric factor for $\mathbf{V}\mathbf{V}^\top + \mathbf{I}$, we then have that $\mathbf{D}^{-1/2}\mathbf{W}^{-1}$ is a symmetric factor for $\hat{\mathbf{\Sigma}}$. Such a factorization can be found using the work of [4], which showed that by taking

$$\begin{aligned}
\mathbf{A} &= \text{Cholesky}(\mathbf{V}^\top\mathbf{V}), \\
\mathbf{B} &= \text{Cholesky}(\mathbf{V}^\top\mathbf{V} + \mathbf{I}_L), \\
\mathbf{C} &= \mathbf{A}^{-\top}(\mathbf{B} - \mathbf{I}_L)\mathbf{A}^{-1},
\end{aligned} \tag{28}$$

$\mathbf{W} = \mathbf{I}_D + \mathbf{V}\mathbf{C}\mathbf{V}^\top$ is a symmetric factorization for $\mathbf{I}_D + \mathbf{V}\mathbf{V}^\top$. We can then use Woodbury's Identity to avoid taking the inverse in the $D \times D$ space,

$$\mathbf{D}^{-1/2}\left(\mathbf{I}_D + \mathbf{V}\mathbf{C}\mathbf{V}^\top\right)^{-1}\boldsymbol{\epsilon} = \mathbf{D}^{-1/2}\left(\mathbf{I}_D - \mathbf{V}\left(\mathbf{C}^{-1} + \mathbf{V}^\top\mathbf{V}\right)^{-1}\mathbf{V}^\top\right)\boldsymbol{\epsilon} \tag{29}$$

and careful ordering of operations, as above, leads to a $O(DL^2)$ complexity. This subroutine is implemented in Algorithm 3.

# D Diagonal Correction

In this section, we prove that the diagonal of the precision computed by SLANG when $L < D$ is identical to the diagonal computed when $L = D$.

Consider the precision $\hat{\mathbf{\Sigma}}_t^{-1} = \mathbf{D}_t + \mathbf{U}_t\mathbf{U}_t^\top$, where the diagonal and low rank components are updated by (7), (10) and (9). Recall that when $L = D$,

$$\hat{\mathbf{\Sigma}}_t^{-1} = \mathbf{\Sigma}_t^{-1},$$

where $\mathbf{\Sigma}_t^{-1}$ was the precision matrix updated by (2). Assume both methods use the same initial diagonal precision matrix and that they are updated by same sequence of EF matrices $\{\hat{\mathbf{G}}(\boldsymbol{\theta})\}$. Then we will show that at every iteration $t$,

$$\text{diag}\left[\hat{\mathbf{\Sigma}}_t^{-1}\right] = \text{diag}\left[\mathbf{\Sigma}_t^{-1}\right].$$

Figure 6: This figure compares the convergence behavior on Australian for two models: Bayesian logistic regression (left) and Bayesian neural networks (BNN) (right). SLANG(1, 2, 3) refers to $L = 1, 5, 10$ for logistic regression and $L = 8, 16, 32$ for BNN. The mean-field method is a natural-gradient mean-field method for logistic regression (see text) and BBB [7] for the BNN experiment.

**Proof:**

$\hat{\boldsymbol{\Sigma}}_t^{-1}$ and $\boldsymbol{\Sigma}_t^{-1}$ are initialized as the same diagonal matrix, so the claim holds trivially at $t = 0$.

Assume that the claim holds at some iteration $t$. The inductive hypothesis implies

$$\text{diag}\left[\boldsymbol{\Sigma}_t^{-1}\right] = \text{diag}\left[\mathbf{U}_t \mathbf{U}_t^\top\right] + \mathbf{D}_t = \text{diag}\left[\hat{\boldsymbol{\Sigma}}_t^{-1}\right].$$

Applying the update (2) gives the diagonal of $\boldsymbol{\Sigma}_{t+1}^{-1}$ to be

$$\text{diag}\left[\boldsymbol{\Sigma}_{t+1}^{-1}\right] = (1 - \beta)\,\text{diag}\left[\boldsymbol{\Sigma}_t^{-1}\right] + \beta\,\text{diag}\left[\hat{\mathbf{G}}(\boldsymbol{\theta})\right] + \beta\lambda\mathbf{I}$$
$$= (1 - \beta)\,\text{diag}\left[\mathbf{U}_t \mathbf{U}_t^\top\right] + (1 - \beta)\,\mathbf{D}_t + \beta\,\text{diag}\left[\hat{\mathbf{G}}(\boldsymbol{\theta})\right] + \beta\lambda\mathbf{I}$$

The diagonal of the SLANG precision at $t + 1$ is

$$\text{diag}\left[\hat{\boldsymbol{\Sigma}}_{t+1}^{-1}\right] = \text{diag}\left[\mathbf{U}_{t+1} \mathbf{U}_{t+1}^\top\right] + \mathbf{D}_{t+1}$$
$$= \text{diag}\left[\mathbf{U}_{t+1} \mathbf{U}_{t+1}^\top\right] + (1 - \beta)\mathbf{D}_t + \beta\lambda\mathbf{I} + \Delta_D$$
$$= (1 - \beta)\,\text{diag}\left[\mathbf{U}_t \mathbf{U}_t^\top\right] + (1 - \beta)\mathbf{D}_t + \beta\,\text{diag}\left[\hat{\mathbf{G}}(\boldsymbol{\theta})\right] + \beta\lambda\mathbf{I},$$

where the last line is obtained by expanding $\Delta_D$ and canceling the $\text{diag}\left[\mathbf{U}_{t+1} \mathbf{U}_{t+1}^\top\right]$ terms. This completes the proof. In practice, the diagonal of the update might differ because the two methods might update $\boldsymbol{\mu}$ differently. Nevertheless, the above results shows a desirable property of SLANG.

# E   Details for Experiments on Bayesian Logistic Regression

We present additional results for SLANG on logistic regression and then provide algorithmic details for all logistic regression experiments.

## E.1   Additional Results

Additional convergence results for logistic regression are provided in Figure 6, which shows the behavior of SLANG on the Australian dataset. These results are from the same experiments as those presented in Figure 4.

Figure 7 shows qualitative comparisons of posterior means, variances, and covariances for the Australian, Breast Cancer, and a1a datasets, similar to Figure 1. These results resemble those for USPS, where the mean-field method (MF Exact) displays "trend-reversal" for the marginal covariances when compared to the Full-Gaussian Exact method. In comparison, SLANG gives a

good approximation of the ground-truth Full-Gaussian covariance approximation for Australian and Breast Cancer. For the a1a dataset, SLANG with L = 10 fails to learn the covariance structure and shows mixed results on the marginal variances. We believe that this is because the dimensionality of a1a is quite large ($D = 1,605$). We expect SLANG to improve when L is sufficiently increased.

Tables 6 and 7 are more detailed versions of Table 1. The tables are split into baselines and SLANG to improve readability. Table 7 also reports values for L = 2, which are not reported in Table 1 due to space constraints.

(a) Australian-scale

(b) Breast cancer

(c) a1a

| Dataset | MF | SLANG-1 | SLANG-5 | SLANG-10 | Full-Gaussian |
|---|---|---|---|---|---|
| Australian | $19.94, 0.04$ | $10.02, 1.26$ | $13.92, 2.82$ | $18.49, 6.98$ | $24.08, 56.93$ |
| Breast cancer | $4.21, 0.12$ | $4.47, 1.17$ | $4.44, 1.72$ | $4.41, 1.75$ | $4.26, 1.54$ |
| A1A | $-2.13, 0.01$ | $-2.04, 0.16$ | $-2.13, 0.25$ | $-2.17, 0.37$ | $-2.11, 1.37$ |
| USPS 3 vs. 5 | $2.28, 0.03$ | $2.17, 0.67$ | $2.06, 0.95$ | $1.98, 1.38$ | $1.80, 2.09$ |

(d) Table of (mean, variance) for the bias term

Figure 7: Comparison of the posterior approximations of SLANG, full-Gaussian and Mean-Field (MF) methods. The figures on the left compare the structure of the off-diagonal covariance and the figures on the right compare the means and diagonal variances. While the means are closely matched for all methods, the MF approximation underestimates the variances on all three datasets. Note that the diagonal of the covariance is not included in the covariance plot on the left, and the bias term is only shown in the last table - as the off-diagonal, diagonal and bias covariances are of different magnitude, a single scale would make comparison difficult.

Table 6: Comparison of SLANG to many mean-field and full-Gaussian methods. Results for mean-field and full-Gaussian methods are shown in this table, while results for SLANG are shown in Table 7 due to space constraints. We see that SLANG with $L = 1$ shows better performance than mean-field methods. It is also quite close to the performance of full-Gaussian method, except in a1a. We expect SLANG to do better on a1a if we increase the rank further.

| Datasets | Metrics | Mean-Field Methods | | | Full-Gaussian | | |
|---|---|---|---|---|---|---|---|
| | | EF | Hessian | Exact | EF | Hessian | Exact |
| Australian | ELBO | $0.6139 \pm 0.0059$ | $0.6125 \pm 0.0059$ | $0.5933 \pm 0.0058$ | $0.5601 \pm 0.0059$ | $0.5583 \pm 0.0059$ | $0.5589 \pm 0.0059$ |
| | NLL | $0.3480 \pm 0.0069$ | $0.3472 \pm 0.0068$ | $0.3413 \pm 0.0072$ | $0.3396 \pm 0.0072$ | $0.3386 \pm 0.0072$ | $0.3377 \pm 0.0069$ |
| | KL $(\times 10^4)$ | $2.2398 \pm 0.3459$ | $2.0301 \pm 0.3146$ | $0.1946 \pm 0.0214$ | $0.0001 \pm 0.0000$ | $0.0000 \pm 0.0000$ | $0.0000 \pm 0.0000$ |
| Breast Cancer | ELBO | $0.1217 \pm 0.0028$ | $0.1208 \pm 0.0028$ | $0.1205 \pm 0.0028$ | $0.1107 \pm 0.0028$ | $0.1086 \pm 0.0029$ | $0.1087 \pm 0.0029$ |
| | NLL | $0.0950 \pm 0.0024$ | $0.0943 \pm 0.0023$ | $0.0937 \pm 0.0024$ | $0.0920 \pm 0.0023$ | $0.0912 \pm 0.0023$ | $0.0912 \pm 0.0024$ |
| | KL | $8.0188 \pm 0.2540$ | $9.0706 \pm 0.1750$ | $7.7713 \pm 0.1173$ | $0.6373 \pm 0.0221$ | $0.0017 \pm 0.0003$ | $0.0000 \pm 0.0000$ |
| a1a | ELBO | $0.3838 \pm 0.0000$ | $0.3833 \pm 0.0000$ | $0.3828 \pm 0.0000$ | $0.3686 \pm 0.0000$ | $0.3678 \pm 0.0000$ | $0.3679 \pm 0.0000$ |
| | NLL | $0.3390 \pm 0.0000$ | $0.3389 \pm 0.0000$ | $0.3385 \pm 0.0000$ | $0.3386 \pm 0.0000$ | $0.3385 \pm 0.0000$ | $0.3386 \pm 0.0000$ |
| | KL $(\times 10^2)$ | $2.5896 \pm 0.0000$ | $2.2082 \pm 0.0000$ | $1.2946 \pm 0.0000$ | $0.0141 \pm 0.0000$ | $0.0001 \pm 0.0000$ | $0.0000 \pm 0.0000$ |
| USPS (3vs5) | ELBO | $0.2679 \pm 0.0029$ | $0.2675 \pm 0.0029$ | $0.2672 \pm 0.0028$ | $0.1886 \pm 0.0022$ | $0.1860 \pm 0.0022$ | $0.1860 \pm 0.0022$ |
| | NLL | $0.1390 \pm 0.0020$ | $0.1388 \pm 0.0020$ | $0.1383 \pm 0.0020$ | $0.1309 \pm 0.0020$ | $0.1300 \pm 0.0020$ | $0.1301 \pm 0.0020$ |
| | KL $(\times 10^1)$ | $7.6836 \pm 0.1485$ | $7.1878 \pm 0.0978$ | $7.0834 \pm 0.0893$ | $0.1797 \pm 0.0022$ | $0.0012 \pm 0.0002$ | $0.0000 \pm 0.0000$ |

Table 7: Comparison of SLANG to many mean-field and full-Gaussian methods. The performance of SLANG for different L is shown in this table, while results for mean-field and full-Gaussian methods are reported in Table 6.

| Datasets | Metrics | SLANG | | | |
|---|---|---|---|---|---|
| | | L = 1 | L = 2 | L = 5 | L = 10 |
| Australian | ELBO | $0.5744 \pm 0.0055$ | $0.5743 \pm 0.0055$ | $0.5690 \pm 0.0056$ | $0.5659 \pm 0.0058$ |
| | NLL | $0.3415 \pm 0.0065$ | $0.3416 \pm 0.0065$ | $0.3392 \pm 0.0065$ | $0.3382 \pm 0.0066$ |
| | KL $(\times 10^4)$ | $0.0332 \pm 0.0068$ | $0.0313 \pm 0.0067$ | $0.0084 \pm 0.0020$ | $0.0017 \pm 0.0003$ |
| Breast Cancer | ELBO | $0.1117 \pm 0.0029$ | $0.1111 \pm 0.0028$ | $0.1114 \pm 0.0028$ | $0.1107 \pm 0.0028$ |
| | NLL | $0.0921 \pm 0.0023$ | $0.0918 \pm 0.0023$ | $0.0919 \pm 0.0023$ | $0.0920 \pm 0.0023$ |
| | KL | $0.9112 \pm 0.0177$ | $0.7560 \pm 0.0290$ | $0.8418 \pm 0.0240$ | $0.6376 \pm 0.0222$ |
| a1a | ELBO | $0.3766 \pm 0.0000$ | $0.3759 \pm 0.0000$ | $0.3744 \pm 0.0000$ | $0.3732 \pm 0.0000$ |
| | NLL | $0.3386 \pm 0.0000$ | $0.3385 \pm 0.0000$ | $0.3386 \pm 0.0000$ | $0.3386 \pm 0.0000$ |
| | KL $(\times 10^2)$ | $0.3051 \pm 0.0000$ | $0.2490 \pm 0.0000$ | $0.1731 \pm 0.0000$ | $0.1179 \pm 0.0000$ |
| USPS (3vs5) | ELBO | $0.2096 \pm 0.0025$ | $0.2059 \pm 0.0024$ | $0.1979 \pm 0.0024$ | $0.1929 \pm 0.0023$ |
| | NLL | $0.1325 \pm 0.0019$ | $0.1325 \pm 0.0019$ | $0.1317 \pm 0.0019$ | $0.1314 \pm 0.0019$ |
| | KL $(\times 10^1)$ | $1.4924 \pm 0.0199$ | $1.2457 \pm 0.0175$ | $0.7547 \pm 0.0110$ | $0.4481 \pm 0.0058$ |

Table 8: A list of datasets for logistic regression. $N_{\text{Train}}$ is the number of training data. $\lambda$ is the precision of the prior distribution used in our logistic regression experiments.

| Dataset | $N$ | $D$ | $N_{\text{Train}}$ | Prior Precision | $M$ |
|---------|-----|-----|--------------------|-----------------|-----|
| USPS3vs5 | 1,781 | 256 | 884 | $\lambda = 25$ | 64 |
| a1a | 32,561 | 123 | 1,605 | $\lambda = 2.8072$ | 128 |
| Australian-scale | 690 | 14 | 345 | $\lambda = 10^{-5}$ | 32 |
| Breast-cancer-scale | 683 | 10 | 341 | $\lambda = 1.0$ | 32 |

Table 9: Learning rates for the logistic regression convergence experiments in Figures.

| | Mean-Field | | SLANG | | | | Full-Gaussian | |
|---------|--------|--------|---------|---------|---------|----------|--------|--------|
| **Dataset** | **EF** | **Hess.** | **L = 1** | **L = 2** | **L = 5** | **L = 10** | **EF** | **Hess.** |
| Australian | 0.0215 | 0.0215 | 0.0117 | 0.0117 | 0.0117 | 0.0117 | 0.0117 | 0.0117 |
| Breast Cancer | 0.0215 | 0.0215 | 0.0398 | 0.0398 | 0.0398 | 0.0398 | 0.0398 | 0.0398 |
| USPS 3vs5 | 0.0063 | 0.0063 | 0.0117 | 0.0117 | 0.0215 | 0.0398 | 0.0398 | 0.0398 |

### E.2 Algorithmic Details for Logistic Regression Results (Table 1)

Datasets for logistic regression are available at `https://www.csie.ntu.edu.tw/~cjlin/libsvmtools/datasets/binary.html`. We used the model hyper-parameters found by [17] for all datasets except for USPS. All details are given in Table 8, which is reproduced from their paper. We selected a relatively strong prior for USPS to avoid overfitting, but did not search for an optimal precision.

For all datasets except a1a, we ran each method on 20 different 50%-50% training-test splits of the datasets. a1a is pre-split into a training and test set and so we only report values for the provided split. Each method was run for 10,000 epochs. We initially set $\alpha_0 = \beta_0 = 0.05$. We then decayed the learning rates at every iteration as follows:

$$\alpha_t = \beta_t = \frac{\alpha_0}{(1 + t^{0.51})}$$

Using a large number of epochs and slowly decaying the learning rates ensured that the considered methods converged. The number of MC samples used was 12. For each dataset, we used a batch size that was roughly one-tenth of the total training set size. These sizes are shown in Table 8. On all experiments, SLANG used momentum for the mean parameter $\boldsymbol{\mu}$, with the parameter set to $\gamma = 0.9$.

Finally, we used the final covariance matrices learned on the first training-test splits of the datasets to generate Figures 1 and 7.

### E.3 Algorithmic Details for Logistic Regression Convergence Experiment (Figure 4)

We used the same hyperparameters as in the previous logistic regression experiments on the LIBSVM datasets. These are reported in Table 8. We used the following procedure to select the learning rates separately for each method:

1. The learning rates that we considered were:
   $\alpha = \beta \in \{0.0010, 0.0018, 0.0034, 0.0063, 0.0117, 0.0215, 0.0398, 0.0736, 0.1359, 0.2512\}$

2. We ran three restarts with different random seeds on the same split of the data for each potential learning rate. These restarts ran for 5,000 epochs with 12 MC samples and the batch sizes listed in Table 8. We did not use a decay on the learning rate.

3. We visually inspected the mean and variance of the training loss against epochs. For each method, we chose the learning rate assignment that produced the fastest convergence with tolerable variance. Variances were compared across methods to ensure consistency.

The learning rates selected in this manner are reported in Table 9.

To obtain the final convergence results, each method was run with ten different random seeds on the same training-test split of the datasets. We trained for 2,000 epochs on all datasets. The number of MC samples used was 12. Once again, the minibatch sizes listed in Table 8 were used. The learning rates were not decayed.

Figure 8: Results for a synthetic toy data. Each plot shows the predictive distribution of a method along with the data examples shown in blue, except for the first plot in the bottom row which shows the value of the negative ELBO for the last 2,000 iterations. The stars in the convergence plot indicate the selected model for SLANG-1 and SLANG-5.

# F Details for Experiments on Bayesian Neural Networks

## F.1 Algorithmic Details for Regression Curves Experiment

In Figure 8, we qualitatively examine the posterior approximations computed by SLANG for neural network models using a synthetic regression data set. The data was generated from the noisy cubic function

$$x \sim \mathrm{U}[-4, 4] \text{ and } y = x^3 + \epsilon, \ \epsilon \sim \mathcal{N}(0, 9).$$

We show the result of fitting a one-hidden-layer ReLU network with 10 units to 30 data points generated in this manner using SLANG and BBB. During optimization, we the used full dataset and 100 MC samples to compute stochastic gradients. We decayed the learning rates for both the mean and covariance.

All methods properly show increased uncertainty in the function when we move away from the data. In comparison to BBB, SLANG allows for smoother transitions and better representation of uncertainty at the junction of the piece-wise linear functions.

We found that the optimization procedure for SLANG did not necessarily converge on the synthetic regression dataset. Figure 8 shows the value of the ELBO during the last part of the optimization procedure to illustrate the convergence issue. This may be due to the EF matrix $\hat{\mathbf{G}}(\boldsymbol{\theta})$ used in the VOGN update (2). We used the ELBO to select the best model.

## F.2 Algorithmic Details for BNN Convergence Experiment

The datasets for this experiment can be found at `https://www.csie.ntu.edu.tw/~cjlin/libsvmtools/datasets/binary.html`. We used the same 50%-50% training-test splits of the datasets as were used in the logistic regression convergence experiment. The models considered were feed-forward neural networks with a single hidden layer of 50 units. The minibatch sizes were chosen to be the same those given in Table 8. We used isotropic Gaussian priors for all datasets. On all experiments, SLANG used momentum for the mean $\boldsymbol{\mu}$ with the parameter set to $\gamma = 0.9$. Precisions for the prior distributions were chosen by grid search over the following values:

$$\lambda \in \{0.001, 0.01, 0.1, 1, 8, 32, 64, 128, 512\}$$

5-fold cross validation on the each training set was used to estimate the test log-loss; the precisions that resulted in the smallest cross-validated test log-loss were selected. This procedure was conducted

Table 10: Prior precisions for BNN convergence experiments (Figure 4).

| Dataset | Australian | Breast Cancer | USPS 3vs5 |
|---|---|---|---|
| Prior Precision | $\lambda = 8$ | $\lambda = 8$ | $\lambda = 32$ |

separately for SLANG and Bayes by Backprop, but the selected values were found to agree on every dataset. The prior precisions are listed in Table 10.

We used almost the same procedure as in the logistic regression convergence experiments to select the learning rates for SLANG. For Bayes by Backprop, we used the Adam optimizer [20], but carefully chose its learning rate using this procedure as well. The procedure was as follows:

1. The learning rates that we considered were:

$$\alpha = \beta \in \{0.0001, 0.00021544, 0.00046416, 0.001, 0.00215443,$$
$$0.00464159, 0.01, 0.02154435, 0.04641589, 0.1\}$$

2. We ran three restarts with different random seeds for each potential learning rate. These restarts ran for 1,000 epochs with 20 MC samples and the batch sizes listed in 8. Losses were computed using 20 MC samples. We did not decay the learning rates.

3. We visually inspected the mean and variance of the training loss over training epochs. For each method, we chose the learning rate assignment that produced the fastest convergence with tolerable variance. Variances were compared across methods to ensure consistency.

The best learning rate found for SLANG was the same across each dataset and value of $L$: $\alpha = \beta = 0.02154435$. Similarly, Bayes by Backprop performed best on each dataset with: $\alpha = 0.01$.

To obtain the final convergence results, each method was run with ten different random seeds. We trained for 500 epochs on all datasets. The number of MC samples used for training and for model evaluation was 20. Once again, the minibatch sizes listed in Table 8 were used. The learning rates were not decayed.

## F.3 Algorithmic Details for UCI Experiments

Each dataset was split randomly 20 times with $90\%$ of the data in the training set and $10\%$ in the test set. We used the same splits used in [10].

For both SLANG and BBB, we used an isotropic Gaussian prior

$$p(\boldsymbol{\theta}|\lambda) = \mathcal{N}(\boldsymbol{\theta}|0, \lambda^{-1}\mathrm{I}) \tag{30}$$

and a Gaussian likelihood

$$p(y|\boldsymbol{\theta}, \mathbf{x}, \tau) = \mathcal{N}(y|f(\mathbf{x}, \boldsymbol{\theta}), \tau^{-1}), \tag{31}$$

where $f(\mathbf{x}, \boldsymbol{\theta})$ is a neural network parameterized by $\boldsymbol{\theta}$. Following earlier work, we use 30 iterations of Bayesian optimization (BO) to tune $\lambda$ and $\tau$. For each iteration of BO, 5-fold cross-validation was used on the current training set (for one of the 20 random splits) to evaluate the parameter setting. For the optimal setting found by BO, one network was trained on the current training set and evaluated on the current test set. This was repeated for each of the 20 random splits. For each dataset, the final values reported in the table are the mean and standard error from these 20 runs.

For the 5 smallest datasets, we used a mini-batch size of 10 and 4 Monte-Carlo samples during training. For the 3 larger datasets, we used a mini-batch size of 100 and 2 Monte-Carlo samples during training. For all runs we used 120 epochs for the methods to converge.

For BBB, we used the Adam [20] optimizer. The learning rates were set individually for each method and dataset based on the cross-validation performances from an initial random search over learning rates, prior precision and noise precisions. The random search was also used to determine the search spaces for the Bayesian optimization used to tune the prior precision and the noise precision.

## F.4 Algorithmic Details for MNIST Experiments

We fit a Bayesian neural network with two hidden layers, each with 400 hidden units and ReLU activations, to the MNIST dataset. This was done using SLANG with $L \in \{1, 2, 4, 8, 16, 32\}$. During

training, a batch size of 200 was used along with 4 MC samples. The momentum was set to $\gamma = 0.9$. The learning rates $\alpha$ and $\beta$ were initialized to $\alpha_0 = 0.1$ and decayed according to

$$\alpha_t = \beta_t = \frac{\alpha_0}{(1 + t^\omega)},$$

where $\omega$ is the decay rate.

The prior precision $\lambda$, along with the learning rate decay rate $\omega$ was tuned. Denote $\sigma := \sqrt{1/\lambda}$. For each value of $L$, we considered each combination of

$$(-\log \sigma, \omega) \in \{0, 1, 2\} \times \{0.52, 0.54, 0.56, 0.58, 0.60\}.$$

The 60,000 training points for MNIST were split into a training set of 50,000 and a validation set of 10,000. After training for 100 epochs, the best performing configuration, according to validation error, was selected for each value of $L$. The models selected by the tuning procedure were trained further on all 60,000 training points for 300 more epochs. Finally, each model made predictions on the test set. Both during computation of the validation and the test loss, 1,000 MC samples were used.

## Footnotes

[3] https://github.com/facebook/fbpca, https://research.fb.com/fast-randomized-svd/