[Reviews · NeurIPS 2018]

Reviewer 1



UPDATE Thank you for your rebuttal. I maintain my score. -------------- OLD REVIEW Uncertainty estimation is important in bayesian deep learning. The traditional approach is to use diagonal Gaussian posterior approximations. This paper proposes Gaussian posterior approximations where the covariance matrix is approximated using the sum of a diagonal matrix and a low-rank matrix. The paper then outlines an efficient algorithm (with complexity linear in the number of dimensions of the data) that solely depends on gradients of the log likelihood. This is made possible by an approximation of the Hessian that depends on the gradient instead of the second derivatives of the log likelihood. Experiments are conducted on UCI datasets. Comparisons are made against three baselines: Mean Field approximation, Bayes By Backprop [1], and Bayesian Dropout [2]. Clarity: the paper is very well written and easy to follow. Originality and Significance: The approach is novel and the gain in complexity is significant. However I have some reservations on the applicability of the method given the scale of the experiments. Quality: The experiments are rather toy. The datasets are small and low-dimensional. Given the method proposed in this paper has complexity linear in D instead of the usual O(D^3) (where D is the dimensionality of the data), I was expecting experiments on datasets with higher values of D than the ones used in the paper. For example a simple Bayesian Neural Network classification on MNIST would be good (See for example [1]). Furthermore, the quantitative results leave me wondering how useful the method is compared to simply using Bayesian Dropout of [2]. I also would have liked to see some qualitative results on the learned weight uncertainties. See for example Figure 5 of [1]. Minor Comments: (1) Equation 3 is a consequence of two equivalent definitions of Fisher Information matrix. (2) I enjoyed the analyses around the effect of the rank on performance. [1] Weight Uncertainty in Neural Networks. Blundell et al. 2015. [2] Dropout as a bayesian approximation: Representing model uncertainty in deep learning. Gal and Ghahramani, 2016.

Reviewer 2



Update: I have considered the author response. I maintain my assessment. ========================================================== This paper proposed an interesting idea of learning efficient structured multivariate Gaussian variational distributions. Since directly making a mean-field approximation may make the variational distribution too far from the model posterior distribution while making a full Gaussian approximation may be intractable, the authors proposed that, we could approximate the inverse of covariance matrix of the variational distribution as the sum of a diagonal matrix and a low-rank matrix. To make the inference faster, the authors also built a natural gradient update algorithm based on the method proposed in [1]. Strengths: In my opinion, this paper addresses an interesting question and many people in the machine learning community may be interested in it. Variational inference is useful since it is fast but the most common variational distribution family, the mean-field family, could not yield satisfactory approximations in some cases. Hence, the problem that this paper focuses on is interesting. The design of their structured variational approximation is guaranteed to work at least as good as the regular mean-field approximation, and could outperform it in many cases, even if the added low-rank approximation only have rank L=1. Moreover, the paper is well-written and it is easy to follow the ideas. Weakness: It might show the success of the proposed method better if some more experiment results are presented. First, for both the Bayesian Logistic regression experiment and the Bayesian neural network experiment, the authors mentioned that a bigger value of L improved the performances, but they only show the results for L=1 (on the metrics in Table 1 and 2). It will be more interesting if the experiment results for larger L could be shown and be compared to the baselines since for both experiments, there are baseline methods work better than the proposed methods. Though we understand that, for the Bayesian Logistic regression experiment, it is not possible for the SLANG method to outperform the full Gaussian method, but the difference between them might be smaller if L is larger. Second, as shown in the right subfigure in Figure 3, the authors tried to compare the efficiency of SLANG with some baseline methods. It will be great if the authors could also compare the efficiency of their methods with some other second-order variational inference algorithms, such as the Hessian-Free Stochastic Gaussian Variational Inference (HFSGVI) algorithm [2]. In addition to that, it will be good if the authors could provide some theoretical analysis on their method. It will be interesting to see the performance improvements of this method compared to the mean-field approximation. Minor comment: Some sentences are a little bit confusing. For example, the sentence "Our goal is to approximate it to bring the cost down to O(D)" on line 110. We know that the time complexity is O(DL^2+DLS), not O(D). Though we understand L and S are small, and the precise time complexity is also mentioned later. It is still a little bit confusing to have this sentence. Probably it is better to say "bring the cost down to be linear in D". References [1] Mohammad Emtiyaz Khan and Wu Lin. Conjugate-computation variational inference: Converting variational inference in non-conjugate models to inferences in conjugate models. arXiv preprint arXiv:1703.04265, 2017. [2] Fan, K., Wang, Z., Beck, J., Kwok, J., & Heller, K. A. (2015). Fast second order stochastic backpropagation for variational inference. In Advances in Neural Information Processing Systems (pp. 1387-1395).

Reviewer 3



This paper considers a modification of a full Gaussian natural gradient variational approximation scheme suggested previously in the literature. It achieves greater scalability through a low rank plus diagonal approximation to the precision matrix in the variational parameter updates. The authors argue that the ability to capture dependence in the posterior is important in some applications and the approach can be implemented in high dimensions. I think this is a high quality contribution, the manuscript is clearly written and I think the methods described are useful. I have a few questions for the authors. 1. How easy is it to relax the assumption of isotropic Gaussian priors? Is there any prospect to incorporate dependence or non-Gaussian sparse shrinkage priors having Gaussian scale mixture representations? 2. Given the many approximations made, is there anything that can be said theoretically about convergence? 3. The examples consider logistic regression and neural networks with a single hidden layer. This seems at odds with the deep learning title of the manuscript. In the examples in Section 5.2, predictive performance is described for the case of L=1 and this works well. For larger L Figure 3 explores the convergence rate in the optimization and larger L gives faster convergence in terms of the number of iterations. However, for the case of neural networks I wonder whether larger L also means better predictive performance. Relating to a remark at the end of Section 5.1 of the authors, if this is not the case it reflects the variational objective more than their proposed method but it would be nice to see some comment on that issue.